# Effects of Ferroptosis on Male Reproduction

**DOI:** 10.3390/ijms23137139

**Published:** 2022-06-27

**Authors:** Yang Liu, Xuanhong Cao, Chen He, Xinrui Guo, Hui Cai, Aili Aierken, Jinlian Hua, Sha Peng

**Affiliations:** College of Veterinary Medicine, Shaanxi Centre of Stem Cells Engineering & Technology, Northwest A & F University, Yangling, Xianyang 712100, China; 2021050619@nwafu.edu.cn (Y.L.); stylecxh@163.com (X.C.); hcl1996@sina.com (C.H.); 2021055522@nwafu.edu.cn (X.G.); 18346700148@163.com (H.C.); alijan@nwafu.edu.cn (A.A.); jinlianhua@nwsuaf.edu.cn (J.H.)

**Keywords:** ferroptosis, iron metabolism, male reproductive disorders, oxidative stress

## Abstract

Ferroptosis is a relatively novel form of regulated cell death that was discovered in 2012. With the increasing research related to the mechanisms of ferroptosis, previous studies have demonstrated that the inactive of the intracellular antioxidant system and iron overload can result in the accumulation of reactive oxygen species (ROS), which can ultimately cause lipid peroxidation in the various cell types of the body. ROS accumulation can cause sperm damage by attacking the plasma membrane and damaging DNA. Acute ferroptosis causes oxidative damage to sperm DNA and testicular oxidative stress, thereby causing male reproductive dysfunction. This review aims to discuss the metabolic network of ferroptosis, summarize and analyze the relationship between male reproductive diseases caused by iron overload as well as lipid peroxidation, and provide a novel direction for the research and prevention of various male reproductive diseases.

## 1. Introduction

As a transition metal, iron is an indispensable trace element present in almost all living organisms and an essential component of heme. Iron can display a variety of functions in the body. For example, it is involved in the cellular oxygen transport and regulation of metabolic energy supply. Consequently, there are specialized proteins and strictly regulated homeostasis mechanisms in humans as well as other organisms to absorb, transport, store, and export iron to prevent iron deficiency or overload in the body. Insufficient iron can cause iron-deficiency anemia, which has been found to primarily affect children and young women [1]. On the contrary, when iron is overloaded, redox homeostasis has been found to be unbalanced, and ROS production is promoted, thereby resulting in oxidative stress, which in turn may cause ferroptosis, a form of iron-dependent lipid peroxidation cell death.

Reproductive disorders in male mammals have been the focus of research in medicine and animal husbandry. Reproductive disorders in male mammals are primarily caused by defective spermatogenic cells, Sertoli cells (SCs), or Leydig cells. The occurrence of male reproductive diseases is vulnerable to the different environmental, physiological, and genetic factors [2]. A number of previous studies have confirmed that oxidative stress is the primary cause of many male reproductive disorders, and it can lead to significant defects in the sperms. Ferroptosis caused by iron overload has been closely related to oxidative stress. Scientists established that injury of the male reproductive system can result from iron overload a long time ago. They found that the abnormal sperm morphology in rats induced by the lack of magnesium and zinc was due to the increase in iron content and oxygen free radicals [3,4,5]. The mechanism of male reproductive disorders caused by iron overload has been thoroughly explored in different studies. In addition, with the discovery of the novel form of cell death termed ferroptosis in recent years, some male reproductive disorders due to iron metabolism disorders have been linked to this process. This review will introduce and summarize the potential effects of iron metabolism and its associated disorders on the male reproductive system from the perspective of ferroptosis.

## 2. Ferroptosis and Its Associated Mechanisms

### 2.1. Overview of Ferroptosis

In the early 2000s, it was discovered that two small-molecule compounds, namely erastin and RAS-selective lethal 3 (RSL3), can selectively kill cancer cells by nonapoptotic mechanisms [6,7]. Interestingly, it was found that cell death induced by these two compounds was not inhibited by inhibitors of apoptosis, necrosis, and autophagy, but was significantly inhibited by iron chelators and lipid peroxide inhibitors. In 2012, scientists found that the small-molecule inducer erasin can kill RAS-mutant tumor cells, and can effectively upregulate transferrin (TF) content and reduce ferritin, thus causing intracellular iron overload. In addition, excessive consumption of glutathione (GSH) and the inactivation of glutathione peroxidase 4 (GPX4) can eventually lead to ROS-mediated lipid peroxidation and cause cell death. This unique form of cell death when first identified was termed ferroptosis [8]. Ferroptosis is an oxidative, iron-dependent form of cell death that significantly differs from apoptosis, classic necrosis, and autophagy, and is primarily characterized by iron-dependent accumulation of lipid hydroperoxides to lethal levels [9]. The morphological characteristics of ferroptosis include cell membrane rupture and vesiculation, decrease in the number of mitochondria, increase in membrane density, mitochondrial ridge decrease or disappearance, mitochondrial outer membrane rupture, normal nuclear size, and lack of chromatin condensation [10]. Ferroptosis has been found to be intimately associated with the occurrence of several chronic diseases, including tumorigenesis [11], neurodegenerative disorders [12], tissue ischemia/reperfusion injury, cardiovascular disorders [13], renal failure, etc. In addition, it is also related to heat stress in plants [14].

### 2.2. Mechanisms of Ferroptosis

The essence of ferroptosis is the accumulation of membrane lipid peroxides due to the substantial increase in iron content. Lipid peroxides are further decomposed into their active derivatives such as aldehydes and reactive oxygen species, which can destroy the different biological macromolecules such as intracellular proteins, lipids, and nucleic acids and ultimately lead to cell death [15]. The metabolism of cysteine (Cys), polyunsaturated fatty acids (PUFAs), and iron has been also closely related to ferroptosis (Figure 1).

#### 2.2.1. Metabolism of Glutathione and Ferroptosis

Selenium-containing GPXs (1–4) can protect cells from oxidative challenge; GPX4, a key regulator of ferroptosis, can especially catalyze the decomposition of lipid peroxides, which are toxic to cells, by utilizing glutathione (GSH) [16]. The consumption of GSH can lead to an imbalance in antioxidant defense and accumulation of lipid ROS, which can induce ferroptosis. For this reason, ferroptosis can also be induced by depriving cells of the essential GSH precursor Cys, or by blocking the function of the GSH-dependent enzyme GPX4 [17]. System X_c_^−^, as a heterodimer consisting of SLC7A11 and SLC3A2, is a cysteine/glutamate reverse-exchange transporter [18]. System X_c_^−^ can transfer glutamate to the outside and cysteine to the inside. Cysteine is an important component of the GSH synthesis pathway. Thus, the uptake of cysteine by System X_c_^−^ can directly affect the amount of GSH production and then alter the activity of GPX4 [19]. SLC7A11 has been identified as a new target for the tumor suppressor gene p53, which can cause cells to become more sensitive to ferroptosis by inhibiting the expression of SLC7A11 and then reducing the uptake of cysteine [20]. In addition, GPX4 can also be affected by the mevalonate pathway and the transculturation pathway to regulate ferroptosis [21]. Ferroptosis can be induced by RSL3, which can reduce the activity of GPX4 by directly combining with GPX4. The second class of compounds, such as erastin, is a direct inhibitor of System X_c_^−^ and can reduce GPX4 activity by consuming glutathione [22]. A wide variety of antioxidants (such as vitamin E and vitamin C), iron-chelating agents, and ferrostatin-1 can inhibit ferroptosis induced by both RSL3 and erastin.

#### 2.2.2. Metabolism of Polyunsaturated Fatty Acids (PUFAs) and Ferroptosis

In addition to lipid peroxidation, another condition that can cause cell ferroptosis is the presence of polyunsaturated fatty acids (PUFAs). Long-chain PUFAs are an important component of all cell membranes. PUFAs contain easily extractable diallyl hydrogen atoms, which are prone to lipid peroxidation and are necessary to perform ferroptosis. Lipid peroxidation can effectively alter the molecular configuration of PUFAs, destroy the fluidity and stability of the cell membrane structure, increase the permeability of the cell membrane, and cause the cells to become more prone to rupture and death. During the formation of lipid peroxides, an important component of the cell membrane, phosphatidylethanolamine (PE), and a type of PUFA, arachidonic acid (AA), are catalyzed by Acyl-CoA synthetase long-chain family member 4 (ACSL4) and lysophosphatidylcholine acyltransferase 3 (LPCAT3) to form AA-PE [23], which can be then peroxidized by iron-dependent lipoxygenases (LOXs) to form the main executor of ferroptosis, AA-OOH-PE [24]. Moreover, deletion of ACSL4 and LPCAT3 can prevent ferroptosis induced by GPX4 inhibitors RSL3 and ML162. In addition, PUFAs can also be oxidized by a nonenzymatic pathway triggered by ROS [15].

#### 2.2.3. Metabolism of Iron and Ferroptosis

The most typical feature of ferroptosis is iron-dependent lipid peroxidation. Iron in cells exists in two different forms: Fe^2+^ and Fe^3+^. Fe^3+^ is relatively stable and can be used for the storage and transportation of iron. Fe^2+^ can participate in various redox reactions. Ferritin is used to store Fe^3+^ in an inert form that lacks the ability to cause lipid peroxidation. The more ferritin, the more Fe^3+^ is stored, and thus a greater resistance can be developed against iron poisoning. When ferroptosis occurs in cells, it produces an excessive amount of Fe^2+^. Fe^2+^ will react with H_2_O_2_ through the Fenton reaction to produce hydroxyl radicals and Fe^3+^. Hydroxyl radical is an important form of reactive oxygen species. These radicals can kill red blood cells and degrade DNA, cell membranes, and various polysaccharide compounds. Hydroxyl radicals can also promote lipid peroxidation, leading to cell death. In other words, iron can nonenzymatically produce ROS through the Fenton reaction [25]. All of these pathways of lipid peroxidation require the participation of iron [26].

## 3. Iron Metabolism and Its Regulation

As a transition metal with redox activity, iron also plays a pivotal role in many physiological processes, including oxygen transport, DNA biosynthesis, and ATP generation [27]. However, iron functions as a double-edged sword, and iron overload in cells can lead to the formation of free radicals and lipid peroxidation. Consequently, stable iron metabolism is of great importance for the body to perform normal cellular activities [28].

### 3.1. Iron Metabolism in the Body

Iron metabolism mainly includes three distinct stages: absorption, storage, and efflux. There are two mechanisms by which iron can enter cells: Tf-dependent manner and Tf-independent manner. Absorption of nonheme iron in food occurs at the brush border of duodenal enterocytes in a Tf-independent manner (Figure 2). Ferric iron in the diet is reduced by membrane-bound iron reductase and then enters small intestinal epithelial cells through divalent metal transporter 1 (DMT1) [29]. Iron in small intestine epithelial cells is transported out of the cells into the blood circulation by ferroportin (FPN). As the only known protein regulated by hepcidin, FPN is primarily responsible for transporting iron out of cells [30]. Transferred ferrous iron is then converted into trivalent iron by ceruloplasmin (Cp), and is loaded on Tf to be transported to other tissues [31]. Most cells take up and utilize iron in a Tf-dependent manner. Under physiological conditions, trivalent iron is tightly bound to Tf in a 2:1 ratio, resulting in a Tf complex, which can then attach to the high-affinity transferrin receptor 1 (TfR1) on the cell surface [32]. The Tf-TfR1 complex is internalized via clathrin-mediated endocytosis. The acidic environment of the endosome can reduce free trivalent iron to divalent iron, which is then transported into the cytoplasm through DMT1 [33]. The majority of iron is transported into the mitochondria to facilitate the synthesis of heme and Fe-S clusters [26], whereas a small portion of iron is used for the formation of ferritin, which is composed of ferritin heavy chain (FTH) and light chain (FTL) (Figure 2). It can effectively absorb divalent iron in the form of multimers to store iron in the cytoplasm [34]. In addition, ferritin also possesses the ability to scavenge iron-mediated free radicals and prevent iron death resulting from excessive iron. Ceruloplasmin plays an important role in iron release from tissue storage. Scientific evidence seems to indicate that a genetic deficiency in mice leads to iron load [35].

### 3.2. Regulation of Iron Metabolism

Intracellular iron metabolism is tightly regulated by a series of different molecules. The ferritin subunit is composed of heavy and light subunits in varying proportions. An iron-responsive element (IRE) in the 5′ untranslated region (5′-UTR) of the ferritin gene can effectively mediate iron-dependent control of its translation [36]. Iron-regulatory protein 1 and 2 (IRP1 and IRP2) primarily determine whether they can bind to IRE to regulate intracellular iron metabolism according to intracellular iron content. Therefore, IRP1 and IRP2 act as the central regulators of cellular iron metabolism [37]. Similar to the ferritin subunit, a similar motif exists in TfR mRNA 3′-UTR, which can interact with IRP to regulate mRNA degradation [38]. Iron and oxygen homeostasis in vivo has also been reported to be regulated by IRP activity through the regulation of hypoxia-inducible factor 2-α (HIF2-α) translation [39]. In addition, ferritinophagy, which releases iron via the nuclear receptor coactivator 4 (NCOA4)-mediated autophagy signaling pathway, can increase the sensitivity of cells to ferroptosis [40]. In mammals, systemic iron homeostasis is controlled by hepcidin, which circulates in the serum and binds to ferroportin (FPN), thereby stimulating FPN degradation and leading to cellular retention of iron, which can result in an overall decrease in the concentration of serum iron. Although the IRE-IRP system and hepcidin-FPN axis exist as two distinct systems that can regulate iron metabolism, there is a close relationship between them because FPN is primarily regulated by IRP [27].

### 3.3. Iron Plays an Important Role in Spermatogenesis

Spermatogenesis is an iron-dependent process in the male reproductive system because developing male reproductive cells undergo successive divisions, and iron is required for both DNA synthesis and cell growth [41]. Iron plays a key role in nucleic acid and protein synthesis, cellular respiration, proliferation, and differentiation, and thus is closely connected with the process of spermatogenesis [42]. These functions imply that there must be a good iron steady state in the seminiferous tubule (SFT), which is mainly composed of Sertoli cells (SCs) and spermatogonia. SCs are a crucial component of the blood/testis barrier (BTB) and are the only cells in the seminiferous epithelium that come in direct contact with the germ cells. In addition, they also possess the function of secreting mediators to promote sperm production as well as maturation, synthesis, and secretion of androgen-binding proteins and secretion of testicular fluid to promote sperm excretion [43]. Therefore, they play a significant role in regulating spermatogenesis. As “nursing cells”, SCs synthesize and secrete Tf and ceruloplasmin (Cp), which are necessary for the transport of iron through the BTB to developing spermatogenic cells at all the different levels.

The transport of iron in the SFT is divided into two distinct steps. Firstly, iron from exogenous Tf is transported into SCs [44]. Tf is then internalized by receptor-mediated endocytosis and thereafter is transported into endosomes [45]. Meanwhile, exogenous Tf remains bound to the receptor and returns to the cell surface for reuse. SCs can then release excess iron into the SFT gap through FPN. The export of iron from SCs to the basement membrane of the SFT also requires a testicular Tf, which is synthesized in the endoplasmic reticulum of SCs. Iron is then combined with newly synthesized testicular Tf to be uptaken by the spermatogenic cells [46]. The mitochondria of the mature sperm are engulfed by SCs, thus facilitating the recycling of iron from back to the primary spermatocyte through the SCs, and completing an autonomous cycle of iron transport within the SFT. Peritubular myoid cells (PTMs) and SCs are the main locations of cytosolic ferritin storage in the testis. During the process of spermatogenesis, the BTB can regulate the entry/exit of substances and the concentration of various bioactive substances in the tube, to modulate iron homeostasis within the seminiferous tubule from the periphery, thus creating a suitable iron homeostasis environment for the testis and protecting the developing germ cells from adverse effects of iron fluctuations [47].

Iron aids in maintaining sperm motility and energy metabolism and is a key factor in supporting sperm function. Ferritin is an important source of iron during sperm development and provides an additional layer of protection for testicular tissues [42]. Ferritin supports ROS production, and heme found in the active site of ferritin can be oxidized by hydrogen peroxide to release free iron, thereby generating iron-based heme and producing more ROS [48]. Fe and nonheme ferritin are also involved in ejaculation thinning, viscosity, sperm pH, and normal spermatogenesis [49]. The importance of iron in male fertility has been demonstrated in various in vivo and in vitro studies [41]. The quantification of iron in bovine spermatoplasm is positively correlated with sperm motility characteristics [50], and it was found that a low concentration of iron can significantly improve sperm motility under in vitro conditions.

Mitochondrial ferritin (MtF), discovered in 2001, is a new type of ferritin that can specifically target the mitochondria. MtF, with a structure and function similar to cytoplasmic ferritin [51], can insulate excessive iron, thus making it less susceptible to Fenton reactions and thus protecting the mitochondria from oxidative damage. MtF is mainly expressed in the sperms and Leydig cells of the testis, which are essential for spermatogenesis [52]. The storage of iron in Leydig cells can thus provide a reservoir of iron for easy access by SCs [42]. High levels of MtF can lead to decreased ferritin expression and increased transferrin receptor expression in the cytoplasm [53]. The regulation of MtF expression and its application in the treatment of the various male reproductive diseases caused by iron overload can serve as the starting point of related research in the future.

## 4. Relationship between Ferroptosis and Male Reproductive Disorders

Sperms can produce ROS, and this process has been reported significantly accelerated in cases of defective sperm function [54]. Thus, sperms are extremely vulnerable to oxidative damage. At present, only a few male reproductive disorders have been proven to be related to ferroptosis. As related research continues, we speculate that several male reproductive disorders may be directly related to ferroptosis (Figure 3). The basic characteristics of ferroptosis are iron overload and lipid peroxidation. Testicular iron overload, inactivation of the antioxidant enzyme GPX4, and lipid metabolism disorders can cause substantial oxidative death (ferroptosis) of cells and therefore lead to the occurrence of male reproductive disorders. Ferroptosis and decreased GPX4 activity can both lead to lipid peroxidation, which in turn can cause ferroptosis [55].

### 4.1. Effects of Ferroptosis on Cells in Testis

A study found that a high iron diet could lower the level of testosterone in the body, which is mainly because increased iron content in the body can mainly cause testicular endoplasmic reticulum stress and affect mitochondrial protein function. Iron overload causes apoptosis of germ cells and Leydig cells in the testis [56]. Leydig cells can synthesize and secrete male hormone stimulated by the interstitial cytosine hormone (luteinizing hormone) secreted by anterior pituitary basophils. The male hormone can promote the occurrence of sperm and male reproductive organ development. In a separate study, histological observations of the testes of mice exposed to iron and cadmium for a long period of time found that iron overload resulted in a reduction in the height of spermatogonia, tubular diameter, and number of germ cells (spermatogonium, spermatocyte, and sperm) [57]. Sertoli cells also play a crucial role in spermatogenesis. When ferroptosis occurs in the testes, the Sertoli cells are damaged. After injury, the expression of ferritin (FPN) decreased, and ROS accumulation and iron content increased in Sertoli cells. For example, ferroptosis was found to contribute to oxygen-glucose deprivation and reoxygenation (OGD/R)-induced supportive cell damage, while cell death, necrosis, and autophagy were not [58].

During spermatogenesis, the only germ cell that is sensitive to ferroptosis stimulation is the round sperm cell, which is extremely sensitive to lipid peroxidation. Round sperm cells are rich in PUFAs but lack key cytoplasmic antioxidants. When ferroptosis occurs in the testis, the levels of ALOX15, ACSL4, and GPX4 proteins in round sperm cells are significantly changed, while those in pachytene spermatocytes are not. Moreover, ferroptosis inducers do not affect sperm motility. Round sperm cells can also die if they lack iron during development. If ALOX15 and ACSL4 are targeted, the effects of ferroptosis on male reproduction can be alleviated. PD146176, an ALOX15 inhibitor, reduces the sensitivity of round sperm cells to lipid peroxidation and the apoptosis rate of round sperm cells [59].

### 4.2. Testicular Lipid Peroxidation and ROS Accumulation Cause Male Reproductive Dysfunction

Intracellular iron overload and failure of antioxidant systems can lead to the excessive accumulation of ROS, which can in turn cause lipid peroxidation. Oxidative stress injury refers to the process in which the balance between ROS generation and ROS scavenging ability of the antioxidant enzyme system is hampered, and ROS accumulation increases, thereby causing oxidative damage to the body. There is growing evidence to implicate a strong link between ferroptosis and oxidative stress. Many diseases mediated by oxidative stress (e.g., cancer, Alzheimer’s disease, etc.) have also been linked to ferroptosis. Oxidative stress can increase ROS concentration and decrease antioxidant capacity. Oxidative stress has been closely related to male reproduction. Oxidative stress can negatively affect sperm quantity, quality, and function by promoting lipid peroxidation, mitochondrial dysfunction, DNA damage, and apoptosis. Under oxidative stress, phospholipids and cholesterol esters containing PUFAs in the cell membranes and lipoproteins can be easily oxidized by the process of lipid peroxidation induced by the free radicals and generate various oxidation products. Ferroptosis is characterized by the destruction of the phospholipid molecules containing unsaturated fatty acids on the membrane or organelle membrane by peroxidation after the inactivation of the intracellular reduction system, thereby resulting in the rupture of the membranes. Mammalian sperm membranes are rich in PUFAs, which are susceptible to lipid peroxidation. Therefore, the abundance of PUFAs can effectively determine the degree of lipid peroxidation that occurs in cells [9]. For instance, arachidonate 15-lipoxygenase (ALOX15), which catalyzes lipid peroxidation, is upregulated in response to oxidative stress in round sperms of the mouse testis treated with RSL3 (inhibitor of GPX4). In addition, during the round stage of sperm cell development, ACSL4’s action can make it sensitive to the process of ferroptosis, which is necessary for the formation of lipid peroxides [59].

Oxidative stress and lipid peroxidation can also adversely affect sperm quality and fertilization by damaging the sperm plasma membrane and DNA. It has been established that low levels of ROS can promote sperm fertilization, but excessive ROS can damage sperm/egg fusion, sperm motility, and DNA integrity. Human sperms are capable of producing ROS, and ROS activity increases significantly in the case of male infertility [60,61]. When NADPH increases to excessive levels, it can stimulate the activity of NADPH oxidase in the plasma membrane of sperm, thereby resulting in the production of excessive ROS. ROS are produced at the mitochondrial level by NADPH-dependent oxidoreductases. When the sperm oxidase system produces an excessive amount of ROS, it can attack the polyunsaturated fatty acids present in high concentrations in the sperm plasma membrane, thereby leading to lipid peroxidation [62,63]. Peroxidation damage to the sperm plasma membrane can result in substantial loss of membrane fluidity and integrity, and thus the sperms lose their ability to participate in fertilization-related membrane fusion events [63,64]. In addition to peroxidation damage to the sperm plasma membrane, ROS can also target DNA and induce human sperm chain breakage and oxidation group damage [65]. ROS has been related to sperm DNA damage, which is positively linked with oxidative stress and negatively correlated with embryo quality and fertilization rate (Figure 3). Sperm DNA damage can significantly reduce the sperm fertilization ability. In the absence of spermatic fluid, sperms are more likely to be exposed to oxidative stress, which can in turn cause DNA strands to break. Sun et al. [66] observed that there was a negative correlation between DNA fragments and sperm quality, and DNA damage could significantly affect sperm density, motility, and morphology. Sperm DNA damage can mainly lead to male infertility through apoptosis during spermatogenesis oxygen free radicals during transport and chromatin abnormality during packaging. However, whether sperm with DNA oxidative damage can still fertilize oocytes apparently depends on the relative rate at which the individual components of DNA integrity and sperm function are lost after being exposed to oxidative stress [67].

### 4.3. Reduced Testosterone Levels Caused by Iron Overload in the Testis

Iron overload can also promote lipid peroxidation and accumulation of ROS and induce ferroptosis. Profoundly low testosterone levels are usually associated with severe iron overload. For instance, a study found that patients with beta-thalassemia major (β-TM), a disorder of hereditary chronic anemia, usually have hypogonadism and defective spermatogenesis. One of the main reasons attributed to this is the accumulation of iron in the pituitary gland and testicles [68,69]. Androgen deficiency observed in rat obesity models has also been found to be caused by elevated iron and hepcidin levels in the testis [56]. On the contrary, the reduction of the male reproductive capacity caused by iron overload is related to oxidative stress. Iron overload tends to trigger Fenton chemistry, leading to cellular oxidative stress and the production of highly reactive free radicals. Scientists have reported that acute iron intoxication can lead to significant oxidative damage to sperm DNA and induce oxidative stress in the testis [70]. Oxidative stress was found to occur when the testis was treated with chloramine, accompanied by increased levels of ferrous and total iron in the testis. Iron oxide nanoparticles (NPs) are commonly utilized for biomedical applications and have been reported to easily pass through the BTB and enter the testis, causing iron accumulation and testicular tissue damage, thereby resulting in oxidative stress [71]. At the same time, decreased Tf and TfR levels have been observed in the seminal plasma from idiopathic azoospermia (IA) patients, which can also be attributed to the upregulation of IRP1 and HIF-1α, leading to dysregulation of DMT1+IRE in the IA testis [72]. The pituitary gland, on the other hand, is particularly sensitive to iron, and iron overload can also lead to severe impairment of the hypothalamus/pituitary/gonadal axis, thus causing hypogonadism and low testosterone levels [73].

Environmental pollutants (such as heavy metals in transportation) can also damage the male reproductive system. Arsenic is a recognized male reproductive toxicant that exists ubiquitously in the environment [74,75]. Arsenite can also cause substantial pathological changes in the mouse testis and significantly reduce the number of sperms. It can also trigger oxidative stress of spermatocytes GC-2 in vitro, thus leading to ferroptosis of the cells [76]. Testicular ischemia/reperfusion (I/R) injury, a common pathophysiological process that occurs with testicular torsion, can also induce the cell death of SCs. It has recently been established that the mechanism of TM4 cell death induced by I/R stress is ferroptosis [58]. Neither lipid peroxidation nor accumulation of ROS in the cells can be inseparable from the role of iron. A number of previous studies have reported that the toxic effect of cadmiumon testis can be modified by tissue concentration of iron [77]. However, when mice were treated with iron and cadmium simultaneously, it was found to increase testicular lipid peroxidation and deplete the concentrations of intratesticular testosterone, cholesterol, and glutathione [57], which may induce ferroptosis cell death in the testis. PM2.5 in the environment can cause dysfunction of male Sertoli cells. Moreover, in vivo and in vitro studies have found that PM2.5 can primarily cause ROS accumulation and iron overload, which can then lead to iron death [78]. Recently, it has been found that the mechanism of di-(2-ethylhexyl) phthalate to cause male reproductive sterility is also iron death induced by testis [79].

### 4.4. Enzymes and Genes Associated with Ferroptosis Play a Role in Spermatogenesis

Iron overload causes ferroptosis-related gene expression and enzyme inactivation, which may result in the dysfunction of the male reproductive system [80]. GPX4 regulates ferroptosis by the cells from harmful lipid peroxidation and maintaining the homeostasis of the membrane lipid bilayer. Ferroptosis can be induced either by direct inhibition of GPX4 or by indirect depletion of glutathione and inhibition of GPX4 activity. Interestingly, direct inhibition of GPX4 can induce substantial iron toxicity. GPX4 in the testis can protect mammalian sperms against spontaneous lipid peroxidation [81,82]. It is noteworthy that GPX4 expression is relatively much higher in the testicles and fat as compared to other tissues. GPX4 not only inhibits lipid peroxidation, but also plays a key role in the process of spermatogenesis. In 2001, Imai et al. reported that GPX4 expression in sperm was significantly reduced in 30% of infertile men diagnosed with oligoasthenospermia. In 2009, they found that the depletion of GPX4 in spermatocytes can cause a marked decline in sperm concentration and motility in mice. Mitochondrial GPX4 (mGPx4) is the main GPX4 product in male reproductive cells. Selenium also plays an important role in the male reproductive system, and its mechanism is related to mGPX4.

Nuclear factor E2-related factor 2 (Nrf2) is a key transcription factor involved in the regulation of antioxidant response and can induce downstream gene expression by binding to multiple antioxidant gene promoter regions and antioxidant response elements (ARE). Transcriptional activation of Nrf2 has been associated with iron resistance death [52]. A number of studies have indicated that the decrease in Nrf2 expression in sperm is related to oligospermia. Moreover, it has been found in mice that when Nrf2 is knocked down, sperm quality decreases significantly in an age-associated manner. Interestingly, by inhibiting ferroptosis, mice oligospermatosis induced by busulfan was reversed, and the protein expression of Nrf2, GPX4, and ferroportin1 (FPN1) induced by busulfan was markedly inhibited. In addition, the use of Nrf2 agonist sulforaphane was found to upregulate the expression of GPX4 as well as FPN1 proteins and substantially improve sperm concentration and motility [83].

One study showed that busulfan induced oligozoospermia in mice by inhibiting Nrf2 expression and subsequently regulating Nrf2 downstream proteins GPX4 and FPN1. Thus, iron appears to be the main hub linking cell death with redox and metabolism. Overall, ferroptosis has been closely related to male reproductive disorders, especially the death of the different cells in the testis caused due to oxidative damage.

### 4.5. Male Reproductive Diseases Should Be Treated by Inhibiting Ferroptosis

Inhibition of iron droop by attenuating the formation of lipid peroxidation products or reducing intracellular iron accumulation might provide protective strategies or treatments for the various related male reproduction. On the one hand, male reproductive disorders can be alleviated through administration of antioxidants. Antioxidants are defined as chemicals that can combat oxidative stress by reducing ROS concentrations or directly reversing oxidative damage. Antioxidants can be broadly divided into small molecules and antioxidant enzymes.

The small molecules are subdivided into fat-soluble (e.g., vitamin E) and water-soluble (e.g., vitamin C) [2]. Vitamin E (α-tocopherol) can protect against polyunsaturated fatty acid (PUFA) peroxidation and prevent the cell damage caused by iron overload [84]. They can protect developing sperm from oxidative attack and DNA damage, which can significantly improve sperm morphology and motility. Vitamin E can restore testicular tissue damage in tissue-specific GPx4 KO mice. Vitamin C (ascorbic acid) can participate in various redox reactions as a direct reducing agent. N-acetylcysteine (NAC) is a widely used antioxidant in metal poisoning [85]. NAC can alleviate oxidative stress by reducing ROS production caused by metal toxicity. Polydatin (PD), the main component of resveratrol, can reduce lipid peroxidation and relieve oxidative stress [86]. PD has also been reported to protect against arsenic-induced testicular damage. In addition, other small molecules that can maintain sperm motility by counteracting lipid peroxidation have also been reported. For instance, taurine is an amino acid that can counteract lipid peroxidation and enhance cellular antioxidant defenses against inflammation. Taurine maintains sperm motility by scavenging ROS and inhibiting lipid peroxidation. Other antioxidants of note include glutathione and carotenoids [87]. Se-GSH-Px and glutathione reductase (GSH-RED) both act as antioxidant enzymes to prevent lipid peroxidation and maintain sperm motility. The former is an antioxidant enzyme related to selenium, while the latter can regenerate and reduce GSH. In addition to these two enzymes, other antioxidant enzymes include SOD, catalase, and glutathione peroxidase.

Iron overload is a necessary condition for the ferroptosis process, and various iron-chelating agents can effectively inhibit ferroptosis of cells. Deferoxamine is a common iron-chelating agent [88] that reduces free iron levels by inhibiting the translation of ferritin mRNA. Deferoxamine can also ameliorate male-related reproductive complications by suppressing iron overload. Ferroportin-1 acts as a scavenger of lipid ROS and can attenuate the action of iron ions. Ferroportin-1 or deferoxamine mesylate was found to reverse the testicular damage induced by busulfan by inhibiting ferroptosis and increasing sperm concentration. Therefore, inhibition of ferroptosis can be used to eliminate the adverse effects of busulfan on male reproduction. Arsenite can induce apoptosis of iron cells significantly in vitro, and the inhibition of iron statin-1 can alleviate this adverse effect. HO-1 is an antioxidant enzyme that can enhance antioxidant defense and reduce ROS production during heat stress in spermatocytes [89]. However, overactivation of HO-1 increases ROS and lipid peroxidation, both of which can be pharmacologically rescued by the iron ion quenching inhibitor ferroportin-1. Smoking can also cause sperm quality decline; the mechanism is primarily related to iron death, and in vitro experiments have found that ferroportin-1 can inhibit ferroptosis in the spermatic fluid [90]. Curcumin acts as an iron-chelating agent that can prevent iron overload by deactivating the activity of iron-regulatory proteins and inhibiting the expression of hepcidin [91]. Coenzyme Q10 inhibits lipid peroxidation by capturing the free radical intermediates, so iron denaturing inhibitor protein 1 and metabolite tetrahydrobiopterin can significantly block lipid peroxidation and inhibit ferroptosis by promoting the formation of reducing CoQ10.

Lifestyle changes can also markedly reduce DNA oxidative damage and improve sperm integrity by alleviating oxidative stress in the sperm. Therefore, inhibition of iron overload and lipid peroxidation with specific pharmacological agents or drugs may be a potential strategy to treat or prevent ferroptosis-related male reproductive diseases.

## 5. Conclusions

Ferroptosis is an abnormal metabolic process that involves amino acids, lipids, and iron. The metabolism of these substances can play a pivotal role in the regulation of various physiological processes. Therefore, any alteration in metabolism can result in injury to the cells. Ferroptosis is characterized by disturbances in metabolism and redox homeostasis, which involves the interaction of elements between complex metabolic networks rather than independent processes. It is extremely susceptible to oxidative stress, which is attributable to frequent cell division during the process of spermatogenesis. The potential relationship between male animal reproductive disorders and oxidative stress in the testis and sperm has been analyzed for a long time. With the discovery of the concept of ferroptosis, scientists have proven that some male reproductive disorders are directly caused by ferroptosis because of the deposition of iron or ROS in the testis. Acute ferroptosis can cause oxidative damage to sperm DNA and can promote oxidative stress in the testes. Sperms can produce ROS, and this activity is significantly accelerated when sperms are functionally deficient. Low levels of ROS can help the sperms to fertilize, whereas high levels can damage sperm. ROS has been closely related to sperm DNA damage, positively linked to oxidative stress, and negatively associated with embryo quality and fertilization rate. Thus, damage to sperm DNA can reduce its ability to fertilize. Oxidative stress and lipid peroxidation can adversely affect sperm quality and fertilization by damaging the sperm plasma membrane and DNA. However, iron overload can cause severe damage to the hypothalamic/pituitary/gonadal axis, thus leading to hypogonadism and low testosterone levels. Iron aids in maintaining sperm motility and energy metabolism and is a key factor in supporting sperm function.

In animal husbandry, the causes of cell damage in the testis involved in some male reproductive disorders (such as the decline in semen quality and the loss of bull libido), which are due to the different environmental factors (such as heat stress), might also have a close relationship with ferroptosis. Overall, the study of ferroptosis and its regulatory mechanisms will not only provide new targets for the therapy of cancer and neurodegenerative diseases, but can also provide innovative ideas for the research and treatment of the various male reproductive diseases caused by oxidative damage and iron overload. ROS and lipid peroxidation can damage sperm, so novel therapeutic strategies to reduce ROS to normal levels are necessary to maintain optimal cellular functions.

## Figures and Tables

**Figure 1 ijms-23-07139-f001:**
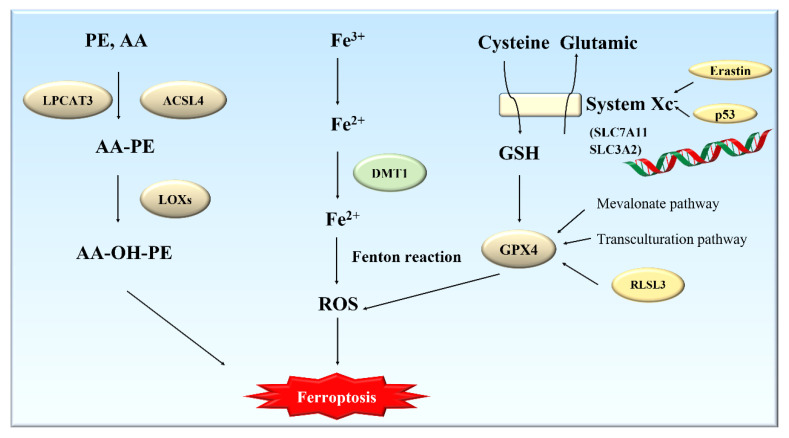
Molecular mechanisms and signaling pathways of ferroptosis.

**Figure 2 ijms-23-07139-f002:**
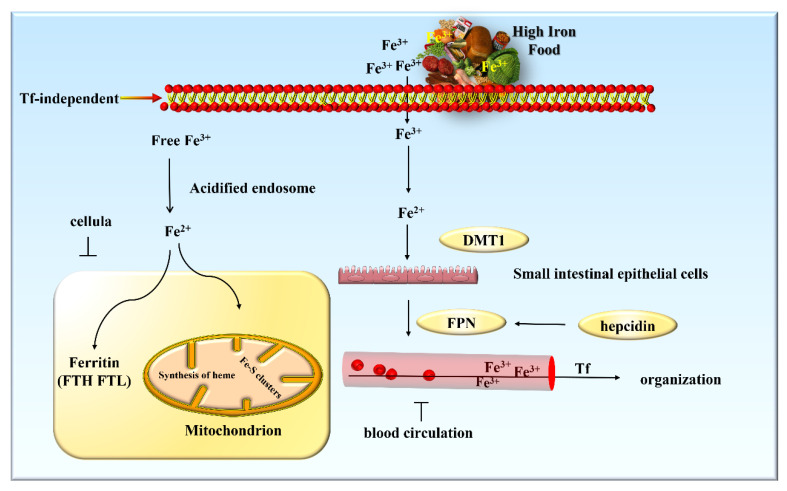
A schematic model describing iron absorption and storage in the body.

**Figure 3 ijms-23-07139-f003:**
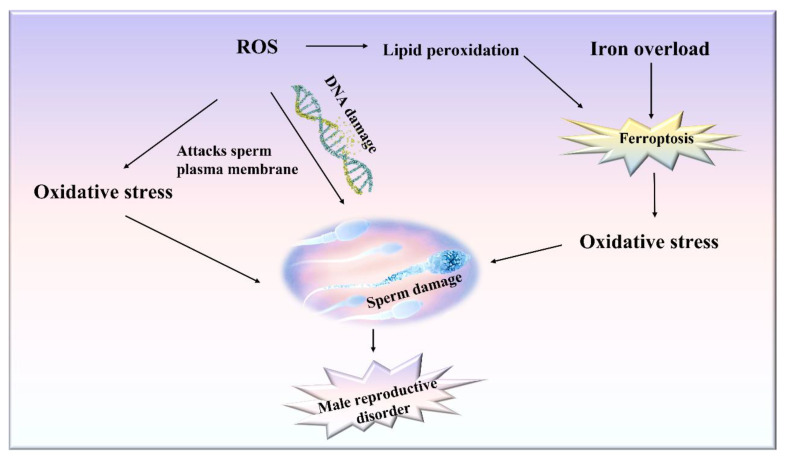
A schematic model describing the potential role of ferroptosis in triggering male spermatogenesis disorders.

## Data Availability

Not applicable.

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
