# Peer review of "Effects of Ferroptosis on Male Reproduction"

_ijms, 2022, doi:10.3390/ijms23137139_

Round 1
Reviewer 1 Report
The review is interesting and well written. However, some minor revision is needed.
There are some missing references within the text:
- in paragraph 2.2;
- in the last sentence of 2.2.2;
- in the last sentence of 3;
- in the last sentence of 3.1;
- in the last sentence of 4;
- in the last sentence of 4.1;
- within the paragraph 4.4.
I found also some minor mistake:
- There is no space between "translation" word and reference number in line 176;
- Check the word "deferoxamine" in line 416;
- Check the space between figures and titles;
- The abbreviation in the table at line 484 are not well aligned.
Author Response
Response to Reviewer 1 Comments
Dear editor and reviewer,
We would like to thank the reviewer and the editor for handling our manuscript and appreciate their constructive comments and suggestions. We have addressed all the concerns raised by you and the reviewers, and have gone to great lengths to improve our manuscript. Please find below the point-to-point response in red. The main corrections in the paper and the response to the reviewer comments are as followings.
Point 1: There are some missing references within the text:
- in paragraph 2.2;
Response 1: Thanks for your comment, we have added the reference to the penultimate sentence of this paragraph.
Point 2: There are some missing references within the text:
- in the last sentence of 2.2.2;
Response 2: Thanks for your advice. We have added the reference to the last sentence of 2.2.2.
Point 3: There are some missing references within the text:
- in the last sentence of 3;
Response 3: Thanks for your suggestion. We have the added reference to the last sentence of 3.
Point 4: There are some missing references within the text:
- in the last sentence of 3.1;
Response 4: Thanks for your suggestion. We have added the reference to the last sentence of 3.1.
Point 5: There are some missing references within the text:
- in the last sentence of 4;
Response 5: Thanks for your advice. We have added the reference to the last sentence of 4.
Point 6: There are some missing references within the text:
- in the last sentence of 4.1;
Response 6: Thanks for your suggestion. We have added reference to the last sentence of 4.2 in revised manuscript.
Point 7: There are some missing references within the text:
- within the paragraph 4.4.
Response 7: Thanks for your suggestion. We have added references to the line 84, 170 and 259 in revised manuscript.
Point 8: - Check the space between figures and titles;
Response 8: Thanks for your suggestion. We have added the space between the title and the figures to the line 441, 447 and 463 in revised manuscript.
Point 9: - The abbreviation in the table at line 484 are not well aligned.
Response 9: Thanks for your proposal. We have aligned it in the line 526 in revised manuscript.
Point 10: - Check the word "deferoxamine" in line 416;
Response 10: Thanks for your advice. We have checked the word "deferoxamine" and corrected it in the line 455 in revised manuscript.
Point 11: There is no space between "translation" word and reference number in line 176;
Response 11: Thanks for your suggestion. We have added the space between "translation" word and reference number in line 183 in revised manuscript.

Reviewer 2 Report
This comprehensive review is valuable for understanding the mechanisms behind ROS and iron involvement in regulation/dysregulation of male fertility.
The reviewer found the authors focused mainly on describing the effect of ROS and iron on the functionality of sperm. However, the authors through the review mentioned their effects on other testicular cells. Therefore, in order to make the review clearer, the reviewer recommend the author to to add subtitles that describe the effects of iron deficiency/overload and ROS and ferroptosis on each cell in the testis such as Sertoli cells, Leydig cells, peritubular cells, spermatogonia, differentiating spermatogonia, round spermatids and sperm.
Author Response
Response to Reviewer 2 Comments
Dear editor and reviewer,
We would like to thank the reviewer and the editor for handling our manuscript and appreciate their constructive comments and suggestions. We have addressed all the concerns raised by you and the reviewers, and have gone to great lengths to improve our manuscript. Please find below the point-to-point response in red. The main corrections in the paper and the response to the reviewer comments are as followings.
Point 1: This comprehensive review is valuable for understanding the mechanisms behind ROS and iron involvement in regulation/dysregulation of male fertility.
The reviewer found the authors focused mainly on describing the effect of ROS and iron on the functionality of sperm. However, the authors through the review mentioned their effects on other testicular cells. Therefore, in order to make the review clearer, the reviewer recommend the author to to add subtitles that describe the effects of iron deficiency/overload and ROS and ferroptosis on each cell in the testis such as Sertoli cells, Leydig cells, peritubular cells, spermatogonia, differentiating spermatogonia, round spermatids and sperm.
Response 1: Thanks for your constructive suggestion. According to the existing literature, we have added a new subtitle named “Effects of ferroptosis on cells in testis” in line 262 in the revised manuscript. And we have described the effects of ferroptosis on cells in the testis.
